# Dysfunctional and Dysregulated Nitric Oxide Synthases in Cardiovascular Disease: Mechanisms and Therapeutic Potential

**DOI:** 10.3390/ijms242015200

**Published:** 2023-10-15

**Authors:** Roman Roy, Joshua Wilcox, Andrew J. Webb, Kevin O’Gallagher

**Affiliations:** 1Cardiovascular Department, King’s College Hospital NHS Foundation Trust, London SE5 9RS, UK; roman.roy@kcl.ac.uk; 2Cardiovascular Department, Guy’s and St. Thomas’ NHS Foundation Trust, London SE1 7EH, UK; joshua.wilcox@kcl.ac.uk; 3Department of Clinical Pharmacology, British Heart Foundation Centre, School of Cardiovascular and Metabolic Medicine and Sciences, King’s College London, London SE1 7EH, UK; andrew.1.webb@kcl.ac.uk; 4British Heart Foundation Centre of Research Excellence, School of Cardiovascular and Metabolic Medicine & Sciences, Faculty of Life Sciences & Medicine, King’s College London, London SE5 9NU, UK

**Keywords:** nitric oxide synthase, endothelial NOS, neuronal NOS, inducible NOS, NOS inhibitor, cardiovascular disease

## Abstract

Nitric oxide (NO) plays an important and diverse signalling role in the cardiovascular system, contributing to the regulation of vascular tone, endothelial function, myocardial function, haemostasis, and thrombosis, amongst many other roles. NO is synthesised through the nitric oxide synthase (NOS)-dependent L-arginine-NO pathway, as well as the nitrate-nitrite-NO pathway. The three isoforms of NOS, namely neuronal (NOS1), inducible (NOS2), and endothelial (NOS3), have different localisation and functions in the human body, and are consequently thought to have differing pathophysiological roles. Furthermore, as we continue to develop a deepened understanding of the different roles of NOS isoforms in disease, the possibility of therapeutically modulating NOS activity has emerged. Indeed, impaired (or dysfunctional), as well as overactive (or dysregulated) NOS activity are attractive therapeutic targets in cardiovascular disease. This review aims to describe recent advances in elucidating the physiological role of NOS isoforms within the cardiovascular system, as well as mechanisms of dysfunctional and dysregulated NOS in cardiovascular disease. We then discuss the modulation of NO and NOS activity as a target in the development of novel cardiovascular therapeutics.

## 1. Introduction

Nitric oxide (NO) plays an important signalling role in multiple organ systems, notably within the cardiovascular system. Whilst originally described as a vasoactive molecule that causes the relaxation of smooth muscles and thereby vasodilatation, NO has also been shown to exert a myriad of other effects, including contributing to endothelial function, myocardial function [1], and neuronal signalling [2].

The biosynthesis of NO in humans is via two pathways: the L-arginine-NO pathway, and the nitrate-nitrite-NO pathway. The L-arginine-NO pathway relies on nitric oxide synthases (NOSs) catalysing the conversion of L-arginine to L-citrulline, yielding NO in the process. Three isoforms of NOS have been described in humans, namely neuronal (nNOS, NOS1), inducible (iNOS, NOS2), and endothelial (eNOS, NOS3). The isoforms of NOS have distinct physiological roles within the human body. Broadly, within the cardiovascular system, eNOS and nNOS regulate arterial blood flow (with each isoform having varying effects dependent on the arterial size) and exert direct myocardial effects. nNOS also has a role in the central nervous system, regulating cerebral blood flow, synaptic plasticity and functional connectivity [3]. iNOS is primarily considered to be physiologically important in the host response to infection and inflammation [4].

An increasing body of evidence has described the physiological role of NOS isoforms and thus the ramifications of impaired or dysfunctional NOS activity, as well as the potential therapeutic benefits of NO-based therapies. Recent evidence has also highlighted the contribution of overactive and dysregulated NOS to a range of cardiovascular diseases, thus making this pathway a promising target for therapeutic interventions. This review will summarise the physiological role of NOS in the cardiovascular system, as well as the pathophysiological effects of dysfunctional and dysregulated NOS and how targeting these pathways may hold therapeutic potential (Figure 1).

## 2. Nitric Oxide Signalling

NO exerts widespread signalling activity, with well-described roles in vascular function, host defence against pathogens, the mediation of inflammation, redox homeostasis, and the regulation of metabolism. These varied roles make NO an important physiological molecule in systems throughout the human body, specifically the cardiovascular, cerebrovascular, respiratory, reproductive system, and many more.

NO primarily acts in a paracrine manner by diffusing into surrounding cells and binding to the haem moiety of soluble guanylate cyclase (sGC) [5]. This leads to an increase in cellular concentration of second messenger cyclic guanylate monophosphate (cGMP), which in turn activates protein kinase G (PKG). PKG subsequently phosphorylates L-type calcium channels, leading to a reduction in intracellular calcium concentration, the relaxation of smooth muscle cells, and vasodilatation. PKG also exerts its actions via the activation of a variety of transcription factors [6].

In addition, NO can exert cGMP-independent effects via the post-translational modification of proteins. S-nitrosylation, the process by which NO covalently binds to cysteine residues of proteins, is the best described mechanism of NO protein modification, and has been linked to various systems and implicated in pathophysiology of many diseases [7,8,9]. Given the large number of proteins that can undergo S-nitrosylation, this provides a mechanism by which NO can exert its many physiological effects. Other, less well described forms of NO-mediated post-translational modification include metal nitrosylation and nitration [7].

## 3. Structure and Function of Nitric Oxide Synthases

As described above, all NOS isoforms convert L-arginine to L-citrulline and NO. This process is haem-dependent and occurs in the presence of co-factors tetrahydrobiopterin (BH_4_), nicotinamide adenine dinucleotide phosphate (NADPH), and O_2_, involving dimerism of the NOS enzyme [10]. All three NOS isoforms depend on calmodulin binding to the enzyme in the transfer of electrons. For eNOS and nNOS, this process occurs in the context of high intracellular calcium concentrations (full activity at 500 nmol/L [11]), whereas iNOS binds to calmodulin very tightly and is therefore fully ‘on’ at resting calcium concentrations. iNOS is therefore considered calcium-independent, enabling it to produce large amounts of cytotoxic NO, which is important in its physiological function in the immune system [12]. NO synthesis by NOS is oxygen-dependent, with rates of synthesis proportional to the oxygen concentration. For nNOS, this occurs over a broad range, with a K_m_O_2_ value (the O_2_ concentration at which the enzyme is at 50% Vmax) of ~400 µM, saturation at ~800 µM, and a Vmax value for NO synthesis of ~2.6 nmol/min, with a similar K_m_O_2_ for NADPH oxidation (~350 μM), with a corresponding Vmax value of ~6.0 nmol/min [13]. For purified iNOS, the K_m_O_2_ is about three-fold lower, ~120 µM, but with a three-fold greater, Vmax > 15 nmol/min, for NADPH consumption [14]. By contrast, eNOS has the weakest activity, with one-sixth and one-tenth the activity of nNOS and iNOS, respectively (due, at least in part, to a shorter hinge element with a unique composition that connects to the FMN module in the reductase domain) but with a much lower K_m_O_2_ of ~4 µM [15]. eNOS and nNOS are constitutively expressed and regulated by transcriptional, post-transcriptional, and post-translational modifications [16], whilst iNOS is induced primarily by gene transcription [17].

## 4. Physiological Role of Nitric Oxide Synthases in the Cardiovascular System

Despite their names, the isoforms of NOS are all expressed in various cell types throughout the body: nNOS is mostly found in neurons, but also in smooth, skeletal, and cardiac muscle; eNOS is typically expressed in endothelial cells, but is also found in smooth muscle, myocardium, and platelets; and iNOS is found in macrophages, smooth muscle, and the liver. Each NOS isoform plays a distinct physiological role within the cardiovascular system, which will be reviewed in detail below.

### 4.1. Endothelial NOS

As its name suggests, eNOS is constitutively expressed in the vascular endothelium, and thus has a crucial role in the regulation of vascular tone and endothelial function. eNOS-derived NO production is regulated by intracellular calcium concentrations, as described above, as well as by shear stress and agonist stimulation by acetylcholine and substance P [11,18,19,20].

Within the vasculature, the primary function of eNOS-derived NO is to vasodilate blood vessels via cGMP-dependent mechanisms as described above. eNOS-derived NO is therefore a significant determinant of tone in multiple vascular beds, contributing to control of cardiovascular haemodynamics.

eNOS-derived NO also plays an important physiological role in vascular protection against thrombosis and atherosclerosis [21]. NO inhibits platelet aggregation by a cGMP-mediated decrease in intracellular calcium flux and resulting in the negative regulation of glycoprotein IIb-IIIa (GP IIb-IIIa), protecting against thrombosis [22]. NO also inhibits leukocyte adhesion to the vascular wall by the inhibition of nuclear factor κB (NFκB), reducing atherosclerosis [23].

NO within the vasculature plays an important but complex role in redox homeostasis. At low levels, eNOS-derived NO acts as an anti-oxidant, reducing reactive oxygen species (ROS) production [22,24,25]. However, at higher concentrations, NO combining with superoxide to form peroxynitrite increases oxidative stress within cells, which has important pathological implications, as discussed below.

In addition to eNOS-derived NO from endothelial cells, eNOS (and to a lesser extent iNOS) contributes to platelet-derived NO [26]. Platelet-derived NO inhibits platelet activation, via cGMP-mediated pathways (decreasing intracellular calcium, inhibiting thromboxane A2 receptor function, and downregulating GP IIb-IIIa receptors [26,27]), but also via cGMP-independent mechanisms (the S-nitrosylation of N-ethylmaleimide-sensitive factor reducing platelet granule exocytosis [28]). Furthermore, through the inhibition of platelet adhesion to endothelial cells and leukocytes as well as promoting the disaggregation of platelets, platelet-derived NO plays an important role in modulating intravascular thrombus formation [26].

eNOS has an important role in the development of and protection against endothelial dysfunction, a state of elevated oxidative stress and inflammation within the endothelium that can lead to the abnormal vasodilatation and vasoconstriction of the vasculature. Endothelial dysfunction has been implicated in a multitude of cardiovascular diseases and diseases of other organ systems [21,29,30,31]. The role of eNOS and NO in endothelial dysfunction is complex; both a decrease and increase in eNOS function have been implicated in endothelial dysfunction. A large number of pre-clinical studies have illustrated the cardioprotective effects of eNOS and thereby its physiological role within the cardiovascular system. eNOS deficiency has been shown to cause a predisposition to atherosclerosis and endothelial dysfunction [32,33], coronary artery disease [32], aortic aneurysm and dissection [32], and hypertension [30]. In addition to its expression in the endothelium, eNOS has more recently been localised to adipocytes and the endothelial cells of perivascular adipose tissue, where it is thought to play a vasculoprotective role [34].

Finally, eNOS is also expressed constitutively in cardiomyocytes, where it primarily localises to the caveolae and plays an important role in mechanoregulation [35]. The effects of NO signalling on cardiac function are complex, depending on the balance in eNOS and nNOS activity within cardiomyocytes and endothelial cells, and also on the influence of β-adrenergic and cholinergic agonism [36]. Within the myocardium, eNOS and nNOS are expressed in different subcellular locations and are understood to have distinct physiological roles [1,37,38]. eNOS activity within the cardiomyocyte is regulated by β-adrenergic and cholinergic drive, and is thought to contribute to net negative chronotropic and inotropic, and positive lusitropic effects [24,35,38]. It is also implicated in the Anrep effect, which describes the positive inotropic effect of myocardial stretch caused by an increased afterload [35]. Beyond its acute effects on cardiac contractility and function, eNOS is also implicated in longer-term cardiac remodelling, acting under physiological circumstances to inhibit pro-hypertrophic stimuli [35].

### 4.2. Neuronal NOS

As implied by its name, nNOS is primarily located in the neurons of the central nervous system (CNS), but is also found in peripheral nitrergic nerves, endothelium, cardiac and skeletal muscle and other cell types. nNOS exerts important haemodynamic effects within the cardiovascular system. The isoform has been localised in the endothelium and/or smooth muscle of multiple vascular beds, including the coronary arteries [39], carotid arteries [40], aorta [40,41], kidneys (macula densa, collecting tubules, and neurons) [42] and microvasculature [19], and has been shown to regulate basal blood flow and vascular tone through these beds [18,19,43]. This regulation of vascular tone occurs through direct vasodilatory effects and via parasympathetic nitrergic nerves. Both nNOS and eNOS contribute to the regulation of arterial tone, with nNOS-derived NO regulating basal arterial tone and thereby systemic vascular resistance and blood pressure [19,44], whilst eNOS contributes to responses to changes in flow stimulated by shear stress or agonist stimulation, as described above [19].

nNOS localised to the cardiomyocyte has been shown to affect both inotropy and dromotropy via effects on intracellular calcium handling [45]. Within the heart, nNOS is found in the cardiomyocytes (sarcoplasmic reticulum, mitochondria, and plasma membrane), intrinsic neurons, and coronary arteries [46,47], contrary to the localisation of eNOS primarily within the caveolae of cardiomyocytes. nNOS regulates cardiac inotropy by increasing the sarcoplasmic reticulum Ca^2+^ ATPase (SERCA) reuptake of calcium and by modulating L-type calcium channel activity, overall reducing contractility and promoting lusitropy [38,47,48]. Beyond calcium handling, nNOS has a physiological role in the heart via its effects on ROS within the cardiomyocyte as well as on mitochondrial proteins (via the inhibition of the mitochondrial respiratory chain) [47].

In addition to its cardiovascular effects, nNOS also exerts important effects within the cerebrovascular system. nNOS-derived NO has been implicated in neurovascular coupling (NVC) [49] and dynamic cerebral autoregulation (dCA), two mechanisms contributing to regulation of cerebral blood flow. NVC describes the relationship between neuronal activity and local CNS blood flow, and is a mechanism by which neural activation leads to an increase in cerebral blood flow (otherwise described as functional hyperaemia). dCA refers to the ability to maintain stable cerebral blood flow despite variations in cerebral perfusion pressures. Cerebral blood flow regulation is complex and consists of a multitude of mechanisms, which act to regulate resistance via large conduit arteries (such as internal carotid arteries, vertebral arteries) but also at the neurovascular unit [50].

One such mechanism involves glutamate release at the synapse-activating N-methyl-D-aspartate (NMDA) receptors, leading to an influx of calcium and thereby nNOS activation. nNOS-derived NO then acts as a vasodilator via the sGC-cGMP-PKG pathway [51], increasing cerebral blood flow. It should be noted that additional nNOS-independent mechanisms of NVC have been described [51]. nNOS-derived NO has been shown to play a role in cerebral blood flow regulation in a variety of physiological and pathological settings, including functional neural activation [3], hypercapnia [52], and hypoxia [53].

Finally, at a neuronal level, nNOS contributes to synaptic plasticity and is thereby implicated in the development of memory and learning, but also in the regulation of neurogenesis [54]. nNOS also has an important role in the cardiovascular response to mental stress; healthy patients exposed to mental stress show nNOS-mediated coronary vasodilatation and increased blood flow [55], whilst patients with arterial hypertension show a blunted nNOS response to mental stress [56].

### 4.3. Inducible NOS

Initially purified from animal macrophages, iNOS has since been found in diverse cell types, including hepatocytes, smooth muscle cells, chondrocytes, CNS cells, and cardiac myocytes [57,58]. As opposed to the constitutively expressed eNOS and nNOS, iNOS has its expression upregulated by stimuli, usually proinflammatory cytokines and/or bacterial lipopolysaccharide [4]. Additionally, and in further contrast to the other two isoforms, iNOS exhibits activity at lower intracellular concentrations of the calcium/calmodulin (CaM) complex (due to tighter binding of CaM to the hinge region of its dimer) [57,59], thus locking iNOS in an “always on” position whereby calcium regulation is no longer relevant. This allows for the high-volume localised production of NO, which is physiologically important in an infection-initiated inflammatory response against multiple microbial pathogens (including viral, bacterial, protozoal and fungal infections) [60,61].

However, iNOS upregulation has been touted as both beneficial and detrimental, with the overproduction of NO via iNOS implicated in disease states such as sepsis [4,12], neurodegenerative disease and stroke [62,63], diabetes and obesity-induced insulin resistance [64], pain syndromes [4], and cancer [4]. Within the cardiovascular system, iNOS induced by inflammatory cytokines provides a mechanism by which inflammation contributes to pathophysiological states, such as atherosclerosis (discussed in more detail below) [65,66].

The inhibition of dysregulated iNOS has been attempted, but studies of in vivo treatment have been met with little success, including in septic shock [12,67] and pain [4]. However, early results in neurodegenerative disease, cancer, obesity-induced insulin resistance/diabetes, and lung disease show promise [4,68].

## 5. Dysfunctional NOS in Cardiovascular Disease

Dysfunctional NOS has been implicated in a variety of cardiovascular diseases. eNOS dysfunction in particular, has been demonstrated in various pathophysiological states, including endothelial dysfunction, atherosclerosis [66], arterial hypertension [30], cardiac hypertrophy [38], and heart failure [38]. The mechanism by which dysfunctional NOS contributes to pathophysiology is generally twofold: a reduction in NOS-derived NO, and NOS “uncoupling” leading to an increase in ROS and oxidative stress. Uncoupling is a state in which NOS generates ROS, including superoxide (O_2_^−^) and hydrogen peroxide (H_2_O_2_) [17]. It occurs under conditions of low L-arginine and BH_4_ concentrations, and has previously been linked to the dissociation of the eNOS dimer to monomers [69,70], although this has more recently been disputed [71]. All three isoforms of NOS are prone to uncoupling, although it is most described in eNOS.

### 5.1. Hypertension

An abundance of evidence has linked hypertension to disruptions in NOS and NO signalling, with eNOS dysfunction and uncoupling particularly implicated [30,72,73]. Reduced vascular eNOS activity and subsequently reduced NO bioavailability leads to impaired vasodilatation and increased peripheral vascular resistance. In addition to changes seen in the regulation of vascular tone, both eNOS and nNOS localised to the myocardium are affected in hypertension; whilst eNOS activity is reduced in hypertensive hearts, nNOS-derived NO is increased and thought to play a protective role [74].

### 5.2. Endothelial Dysfunction

Abnormal eNOS function contributes to endothelial dysfunction, an early hallmark of atherosclerosis, which has been implicated in a wide variety of cardiovascular diseases, including hypertension [30], heart failure [31], and coronary artery disease [29], amongst many others. Endothelial dysfunction is characterised by reduced eNOS-derived NO within blood vessels leading to an impairment in endothelium-dependent relaxation [21]. The mechanisms impairing eNOS activity in disease are multiple, including inhibition by oxidised low-density lipoprotein [33,75] and lysophosphatidylcholine [76], as well as endogenous NOS inhibitors, such as asymmetric dimethylarginine (ADMA) [21,77]. In addition to the inhibition of eNOS-derived NO, the uncoupling of eNOS from its cofactor or substrate leads to production of ROS, further contributing to endothelial dysfunction [78]. nNOS has been shown to be protective against endothelial dysfunction and atherosclerosis, with studies demonstrating the development of accelerated atherosclerosis in nNOS knockout mice [79,80].

Elevated concentrations of arginase, which competes with eNOS for the substrate L-arginine, have been associated with endothelial dysfunction in a number of pathological states including hypertension, atherosclerosis and ischaemia/reperfusion injury [81]. Additionally, the S-glutathionylation of eNOS, promoted by a pro-oxidative environment within the endothelial cell, contributes to eNOS uncoupling [82]. However, eNOS is linked to endothelial dysfunction beyond the uncoupling phenomenon; proline-rich tyrosine kinase 2 (PYK2), activated by not only oxidative stress but also by angiotensin II and insulin, phosphorylates eNOS, inhibiting its effect and reducing NO bioavailability [83].

### 5.3. Myocardial Infarction and Ischaemia/Reperfusion Injury

Evidence has supported a cardioprotective role of both eNOS and nNOS after myocardial infarction (MI) and in ischaemia/reperfusion injury (IRI). Early studies showed that eNOS-knockout mice subjected to coronary artery ligation exhibited higher rates of cardiac remodelling, left ventricular dysfunction and death compared to wild-type mice [84]. Similarly, nNOS-knockout mice subjected to MI showed more cardiac remodelling and higher mortality rates than wild-type mice [85,86]. Furthermore, evidence has pointed towards a role for NOS-derived NO in ischaemic preconditioning (IPC); whilst eNOS may contribute to the early phase of IPC [87], iNOS upregulation is thought to contribute to late IPC [88,89].

A significant evidence base has investigated the use of NO donors in the IRI of the myocardium, implicating reduced NO bioavailability in its pathophysiology [90,91]. eNOS has been observed to be protective after an IRI, with eNOS-deficient mice showing exacerbated IRI compared to wild-type mice [92]. Furthermore, mice overexpressing eNOS have shown attenuated IRI, with beneficial effects of eNOS overexpression being reversed by the non-selective NOS inhibitor N^G^-nitro-L-arginine methyl ester (L-NAME) [93].

### 5.4. Heart Failure

NOS dysfunction has been implicated in myocardial remodelling and heart failure [17,94], with early animal and human studies demonstrating that myocardial eNOS expression is decreased in hypertrophic cardiomyopathy [95], ischaemic cardiomyopathy [96], dilated cardiomyopathy [96,97], and diabetic cardiomyopathy [98]. Conversely, iNOS activity is increased in the failing heart and associated with decreased responsiveness to β-adrenergic stimulation [94,96]. Similarly, Nanos-derived NO is increased in patients with dilated cardiomyopathy [97], although the transgenic overexpression of nNOS in mice has been shown to be protective in IRIs [99]. It remains unclear whether this increase in nNOS activity is adaptive or maladaptive, although the translocation of nNOS to the plasma membrane seen under ischaemic conditions and in heart failure [97] has been postulated to limit myocardial remodelling caused by chronic β-adrenergic stimulation, implying an adaptive role [17]. Animal models have underlined the deleterious effects of eNOS dysfunction whilst highlighting the therapeutic potential of eNOS upregulation in HF. The upregulation of eNOS expression and/or activity in mice has been shown to be protective against post-MI left ventricular dysfunction and remodelling [24,100,101,102], oxidative stress [102] and survival [101], as well as infarct size following IRI [99]. In diabetic cardiomyopathy, studies have shown not only downregulated eNOS but also evidence of uncoupled eNOS activity [98,103]. Upregulated iNOS activity and subsequent increased NO and ROS production has also been associated with diabetic cardiomyopathy [98,103].

More recently, dysfunctional or uncoupled eNOS has been implicated in the pathophysiology of heart failure with preserved ejection (HFpEF). The currently accepted pathophysiological model of HFpEF involves co-morbidities driving systemic and arterial inflammation. At the level of the heart, this inflammation drives eNOS uncoupling in the coronary vasculature, resulting in impaired paracrine NO and cGMP signalling [104], with effects on the giant cytoskeletal protein titin, resulting in increased myocardial stiffness and cardiac diastolic dysfunction [105,106]. However, as will be discussed below, the differing effects of various NOS isoforms must be considered, with evidence of a pathological increase in myocardial NOS activity also described in HFpEF.

### 5.5. Therapeutic Potential

Attempts to target the NO signalling pathways in cardiovascular therapeutics are abundant. Efforts have been made to increase the bioavailability of NO using NO donors, as well as targeting downstream molecules in the NO signalling cascade (downstream targets have been reviewed elsewhere [17]). Additionally, the modulation of dysfunctional NOS has been a target of therapies of cardiovascular disease. Indeed, although not their primary mechanism of action, established cardiovascular drugs, such as those targeting the renin–angiotensin system [107], statins [108], calcium channel blockers [109], and β-adrenergic antagonists [110], have been shown to exert some of their cardio- and vasculoprotective effects via eNOS-dependent mechanisms. Attempts at targeting NO signalling are discussed below, with a focus on targeting dysfunctional NOS.

### 5.6. L-Arginine

As discussed above, the endogenous production of NO by NOS enzymes is dependent on the availability of L-arginine. Consequently, supplementation with L-arginine has been proposed as a means to increase eNOS-derived NO and reduce eNOS uncoupling, which could lead to increased NO bioavailability and reduced ROS production. However, clinical trials investigating L-arginine supplementation in a wide variety of cardiovascular diseases have failed to consistently demonstrate benefits on markers of endothelial function and on clinical outcomes, and have even been associated with possible harm [111,112,113].

One explanation for the lack of efficacy of L-arginine may be its relationship with the endogenous NOS inhibitor ADMA, which competes with L-arginine to bind with NOS, thereby reducing the availability of NOS-derived NO. Studies have shown increased levels of ADMA in patients with heart failure, but also conditions causing predisposition to heart failure, including hypertension, coronary artery disease, valvular disease, endothelial dysfunction, and others [114,115]. Furthermore, L-arginine supplementation may augment iNOS activity and associated deleterious effects via the overproduction of NO and ROS. Finally, human studies have shown impaired L-arginine transport in the failing myocardium and in hypertension, suggesting that the supplementation of L-arginine alone in the context of impaired transport may not be sufficient to improve the bioavailability of NOS-derived NO [116].

### 5.7. BH_4_

Under physiological conditions, BH_4_ acts as a cofactor, allowing the NOS enzyme to remain dimerized and thereby facilitating NOS-derived NO production and preventing uncoupling [24]. BH_4_ supplementation has shown promise in preclinical models of atherosclerosis, inhibiting atherogenesis in apolipoprotein E knockout (ApoE-KO) mice fed a high-cholesterol diet [117], and reversing severe atherosclerosis in ApoE-KO mice with partial carotid ligation fed a high-fat diet [118]. Similarly, the oral supplementation of BH_4_ in mice with transverse aortic constriction-induced heart failure led to recoupled eNOS activity and reversed hypertrophy and fibrosis [119]. Interestingly, a 2013 study showed that the co-administration of L-arginine and BH_4_ was able to protect rats and pigs from IRI [120]. In ex vivo human coronary arterioles from patients undergoing coronary artery bypass grafting (CABG), treatment with sepiapterin (a substrate for BH_4_ synthesis) was able to improve endothelium-dependent vasodilatation [121].

Despite promising preclinical results, clinical evidence of benefit from BH_4_ supplementation is limited. One randomised controlled trial (RCT) in patients undergoing CABG showed that oral BH_4_ had no effect on vascular redox state or endothelial function, potentially due to the oxidation of BH_4_ to BH_2_, which lacks eNOS cofactor activity [122]. This limitation of oral supplementation of BH_4_ presents a major obstacle to its therapeutic use. Similarly, a RCT in patients with CADASIL (cerebral autosomal dominant arteriopathy with subcortical infarcts and leukoencephalopathy) showed no improvement in markers of endothelium-dependent vasodilatation with sapropterin treatment (a synthetic BH_4_ analogue) [123].

### 5.8. NO Donors

NO donors, compounds that release NO and NO-related molecules, have been extensively studied for their potential therapeutic benefits in cardiovascular disease. Glyceryl trinitrate, isosorbide nitrates, and nicorandil are three common examples of NO donors used in cardiovascular disease, primarily for angina and acute MI. Similarly, a wealth of evidence has investigated the therapeutic use of dietary nitrate/nitrite in conditions including hypertension, heart failure (with reduced and preserved ejection fraction; HFrEF and HFpEF), as well as ischaemic heart disease [124,125]. These approaches aim to increase the bioavailability of NO in settings where NOS-derived NO may be deficient [126], as described above. Numerous ongoing studies are investigating the role of dietary nitrate and NO donors in various cardiovascular diseases, including in HFrEF, HFpEF, systemic hypertension, pulmonary hypertension, angina, and stroke [127].

Although the delivery of NO donors is primarily via the oral route in clinical practice, they can also be delivered sublingually, intranasally, transdermally, or intravenously. NO itself can be delivered to the lungs through inhalation, such as for the treatment of pulmonary artery hypertension. Sublingual and immediate-release orally absorbed NO donors tend to have a shorter duration of action compared to transdermal and slow-release oral NO donors. Importantly, barriers to therapeutic success for organic nitrates have included side effects, such as headaches and tolerance, requiring drug-free intervals, as well as endothelial dysfunction, which may develop with chronic use [128]. Slow-release oral nitrates are less likely to cause tolerance than fast release preparations. Furthermore, the NEAT-HFpEF trial found a dose-dependent decrease in daily activity levels in patients taking isosorbide mononitrate versus placebo, highlighting the limitations of organic nitrates [129]. Nitrite, on the other hand, carries the possible adverse effects of methaemoglobinaemia and conversion to carcinogenic nitrosamines [128], limiting its clinical use.

### 5.9. NOS Transcriptional Regulators/Enhancers

In an effort to enhance NOS function in disease states characterised by dysfunctional NOS, transcriptional regulators of NOS activity have been developed. Preclinical studies have demonstrated the potential of transcriptional enhancers for eNOS in improving left ventricular remodelling and contractile dysfunction in rats with experimental MI [130], as well as in reducing hypertrophy/fibrosis and improving diastolic function in a rat model of diastolic heart failure [131]. In addition, they have shown efficacy in reducing cardiac remodelling in mice subjected to aortic banding [132], and reducing platelet activation in rats with post-MI heart failure [133], amongst others [134,135]. Despite these promising preclinical results, the translation of these approaches into clinical research and improved outcomes has yet to be achieved.

### 5.10. Gene Therapy

NOS gene therapy has provided a platform for the improvement of NO bioavailability in cardiovascular diseases. For instance, the delivery of the eNOS gene via an adenovirus vector before MI in rats has been shown to be cardioprotective, reducing infarct size and improving contractility and left ventricular diastolic function [136]. Similarly, the adenovirus vector delivery of eNOS after MI protected against myocardial fibrosis and remodelling as well as apoptosis [102].

Furthermore, adenoviral vectors for the expression of dimethylarginine dimethylaminohydrolase (DDAH), an enzyme that degrades ADMA and thereby increases NO, have shown promise in improving vascular function in mouse carotid arteries [137]. This approach may offer a potential strategy to target endothelial dysfunction in a range of disorders. However, further research is needed to evaluate the efficacy and safety of NOS gene therapy in humans.

## 6. Dysregulated NOS in Cardiovascular Disease

### 6.1. Endothelial Dysfunction, Inflammation, and Oxidative Stress

Alongside evidence for dysfunctional eNOS function, preclinical studies have demonstrated that dysregulated or elevated iNOS activity may also contribute to endothelial dysfunction. Investigations showed that an iNOS-specific inhibitor reversed the impaired pressor responsiveness and endothelial function observed in rats with streptozotocin-induced diabetes, indicating that increased iNOS expression may play a role in diabetes-associated endothelial dysfunction [98]. Similarly, in mice with lipopolysaccharide-induced endothelial dysfunction, iNOS protein expression and plasma NO_x_ levels were increased, whereas iNOS knockout mice exhibited no changes in NO_x_ levels [138], supporting a link between iNOS activity and endothelial dysfunction. Furthermore, early work showed elevated macrophage iNOS expression in a rabbit model of post-MI inflammation [139].

Mechanisms linking inflammation and disease states characterised by inflammation to dysregulated iNOS activity are numerous. Inflammatory cytokines, including IL-1β and IFN-γ, induce iNOS activity [140], leading to the generation of ROS and reactive nitrogen species (RNS) [141]. Furthermore, iNOS-derived NO leads to the upregulation of cyclo-oxygenases (COX), a group of enzymes involved in the generation of prostanoids and thereby contributing to a pro-inflammatory state [65]. Finally, beyond iNOS, the activation of the pro-inflammatory cascade will cause the uncoupling of eNOS, further contributing to reduced NO bioavailability and driving oxidative stress within the cell. These effects lead to an unfavourable oxidant versus antioxidant balance, driving apoptosis, contractile dysfunction, and mitochondrial dysfunction amongst other effects [142].

### 6.2. NO and Mitochondria

In addition to endothelial function, NO and NOS have important effects on mitochondrial health. NO, either from NO donors or produced within the mitochondria itself, reversibly inhibits the action of cytochrome C oxidase, the last enzyme in the electron transport chain, thereby modulating mitochondrial oxygen consumption [143,144]. At higher concentrations, however, NO has harmful effects on the mitochondria. NO causes the inhibition of the mitochondrial respiratory chain, the generation of peroxynitrite, the nitration of mitochondrial proteins, and the release of cytochrome C into the cytosol [143,145]. Peroxynitrite will oxidise a wide variety of substrates (including proteins, tyrosine residues, and DNA, amongst others), inhibiting enzymes at multiple sites of the mitochondrial respiratory chain, and leading to cytochrome C release into the cytosol with subsequent signalling, leading to apoptosis [144,146].

Dysregulated NOS and subsequent excess NO production, therefore, will have detrimental effects on mitochondrial and cell function. iNOS, for example, has been associated with metabolic remodelling and cytokine production, contributing to the pro-inflammatory cascades within macrophages [145].

### 6.3. Heart Failure with Preserved Ejection Fraction (HFpEF)

As described above, inflammation and subsequent eNOS uncoupling within the coronary vasculature has been shown to contribute to the pathophysiology of HFpEF [104,105]. This focus on eNOS function, however, ignores the anatomical compartmentalisation and differing cardiac effects of the various NOS isoforms. Indeed, recent evidence has identified that pathological increases in myocardial NOS activity may be implicated in HFpEF pathophysiology. In a study reporting data from both animal models and human myocardial tissue in patients with cardiac diastolic dysfunction, upregulated/dysregulated nNOS function led to the S-nitrosylation of histone deacetylase, an enzyme known to be involved in the regulation of cardiac hypertrophy [147]. Similarly, animal and human data suggest that upregulated iNOS activity leads to the S-nitrosylation of proteins involved in the cardiomyocyte stress response [94]. Furthermore, it remains unclear how nNOS function in the coronary microvasculature affects myocardial function in a paracrine manner, suggesting possible mechanisms via cellular crosstalk.

The complexity of the pathophysiological role of the different NOS isoforms in HFpEF, as well as their anatomical compartmentalisation may explain why treating patients with NO donors is of limited clinical benefit. Targeting therapy to the dysfunction of specific isoforms with isoform specific NOS inhibitors may provide greater success.

Interestingly, sodium-glucose cotransporter 2 (SGLT2) inhibitors, recently shown to reduce cardiovascular death in patients with HFpEF [148], have been shown to improve NOS coupling [149]—further work will elucidate the extent to which clinical benefits of SGLT2 inhibitors depend upon the modulation of NO signalling pathways.

### 6.4. Coronary Microvascular Disease (CMD)

CMD is increasingly recognised as an important cause of angina in patients that were found to have unobstructed coronary arteries via coronary angiography. Recent work has classified CMD into two key endotypes: functional and structural [150]. In the functional endotype, there is elevated resting basal coronary blood flow. Given that nNOS (rather than eNOS) is responsible for the regulation of basal arterial tone, it has been postulated that the raised basal coronary flow seen in the functional endotype may be due to dysregulated nNOS activity, with studies ongoing to investigate this possibility [151].

### 6.5. Ischaemia-Reperfusion Injury in the Heart

NOS inhibition has been proposed as a potential therapy for IRI following acute coronary syndromes. As the myocardium is reperfused following injury, a complex set of mechanisms, including the generation of free radicals, pro-apoptotic and pro-inflammatory signalling cascades, and endothelial dysfunction, contribute to myocardial damage/cell death, arrhythmias, and myocardial stunning [152,153]. The role of NOS isoforms in IRI is complex: nNOS is upregulated in the myocardium following ischaemia [154] and is thought to mediate the protective effects of ischaemic post-conditioning against IRI. iNOS has been implicated in IRI through worse contractile function and increased oxidative stress via the production of peroxynitrite, but has also been suggested to have a protective role via iNOS-derived NO increasing TNF-α- and COX-2-dependent prostanoids, protecting the myocardium [155]. Recently, iNOS expression was seen to be increased in myocardial tissue from autopsies of patients with acute MI [156], although it is unclear if this is an adaptive or maladaptive change.

Early work investigated non-selective NOS inhibitors, such as N^G^-nitro-L-arginine methyl ester (L-NAME) and N^G^-monomethyl-L-arginine (L-NMMA), in rabbit and rat models of IRI, finding improved recovery of mechanical heart function during the reperfusion period with NOS inhibition [157,158] and reduced infarct size [159]. A subsequent work using a selective inhibitor of iNOS dimerization found improved contractile performance and reduced cell death in isolated perfused rat hearts after IRI [160]. A similar study demonstrated that selective iNOS inhibition and iNOS-knockout mice showed reduced apoptosis and infarct size after IRI in chronic β-adrenergic stimulation-induced cardiac damage in mice [161].

To date there are few clinical studies investigating the use of NOS inhibitors in patients with IRI. A group has investigated NOS inhibition in cardiogenic shock complicating acute MI, with initial studies suggesting that L-NMMA and L-NAME led to haemodynamic improvements [162,163]. However, a subsequent placebo-controlled study in 79 patients showed that L-NMMA resulted only in a modest, short-lived increase in mean arterial pressure at 15 min which was not sustained at two hours [164], and a larger RCT was terminated early for futility, finding no difference in mortality in patients with refractory cardiogenic shock after MI [165].

### 6.6. Post-Stroke Reperfusion Injury

In ischaemic stroke, a large proportion of the neurological damage comes not just from the initial ischaemic insult but also from excitotoxicity, i.e., dysregulated glutaminergic and NO signalling that occurs for several hours/days after the initial ischaemia [166,167]. Current established therapies in ischaemic stroke (e.g., tissue plasminogen activator and mechanical thrombectomy) are aimed at achieving the successful reperfusion of the ischaemic territory but play no direct role in diminishing the inevitable excitotoxicity that follows. Therefore, there remains a significant clinical need for effective treatments to reduce the levels of excitotoxicity and minimise infarct volumes and neurological injury following stroke [168].

Animal data suggest that nNOS contributes significantly to excitotoxicity via cellular crosstalk [169,170]. nNOS is bound to the glutaminergic NMDA receptor by postsynaptic density protein 95 (PSD95), creating a death-inducing signalling complex (Figure 2). NMDA receptor activation in the ischaemic penumbra therefore leads to neurotoxic levels of NO production [168]. Inhibiting nNOS protects against glutaminergic excitotoxicity in animal models of non-ischaemic neurological disease, while nNOS knockout mice are resistant to cerebral ischaemia [171,172,173,174].

Attempts have been made to indirectly interrupt nNOS signalling following acute stroke using the drug NA-1, which inhibits PSD95. In a phase II study, NA-1 was shown to reduce the number of ischaemic infarcts versus placebo in patients undergoing endovascular treatment for intracranial aneurysm [175]. However, in a phase III RCT in patients with ischaemic stroke, NA-1 did not improve functional outcome compared to placebo [176].

Beyond acute (within minutes) detrimental effects of increased nNOS activity, iNOS activity has also been seen to be upregulated in the later stages following stroke (within hours), where its effects are also suggested to be neurotoxic [177]. Conversely, eNOS-derived NO, which is acutely upregulated following stroke, is thought to be neuroprotective [177], underlining the importance of a targeted therapeutic approach.

### 6.7. Therapeutic Potential of NOS Inhibitors

To date, no assessment has been made of the effect of directly inhibiting nNOS in ischaemic stroke. This is largely because direct nNOS inhibition has previously been considered an unfavourable approach, based on behavioural effects seen in murine nNOS knock-out models [178]. However, direct nNOS inhibition with the synthetic L-arginine analogue S-methyl-L-thiocitrulline (SMTC) has been employed in a number of human mechanistic studies without adverse effect [3,18,44,49,55,56], suggesting that the acute systemic and local intra-arterial dosing of SMTC is safe. SMTC is 17 times more selective for nNOS in brain tissue than eNOS in vascular endothelium [179] and is also considered to cross the blood–brain barrier, as demonstrated through use of ^11^C-labelled SMTC in rat and primate models [180]. In a human study assessing the effect of nNOS inhibition on cerebral blood flow responses, changes in functional connectivity between different brain regions were identified, strongly suggesting that SMTC crosses the blood–brain barrier to affect neuronal function rather than simply altering perfusion [3].

Although it may seem counter-intuitive to treat a condition that is primarily one of insufficient blood flow (i.e., ischaemic stroke) with a drug that acts to decrease brain perfusion, it is important to consider that (a) nNOS inhibition is being considered as a potential therapy for post-reperfusion excitotoxicity and would only be given once adequate reperfusion had been achieved, and (b) nNOS-derived NO is one of a number of molecules implicated in the regulation of cerebral blood flow and the effect of nNOS inhibition on cerebral blood flow is modest. In addition, as the interventional management of stroke becomes more widespread, the catheterisation of the cerebral arteries may afford the opportunity to deliver more anatomically targeted treatment, thereby improving the delivery of nNOS inhibitors to the ischaemic brain tissue most at risk of excitotoxicity, and limiting any systemic side effects.

Previous work has investigated the use of agmatine, a compound with pleiotropic effects, which include the modulation of NOS function, in ischaemic stroke. Agmatine, an amine synthesised by the decarboxylation of arginine, acts to inhibit nNOS activity [181,182] whilst enhancing eNOS activity [182,183]. Preclinical studies of agmatine use following stroke have shown promise as a neuroprotective strategy [181,184,185].

### 6.8. Safety Concerns with NOS Inhibitors

The safety profile of NOS inhibitors is critical as their use in conditions with evidence of dysregulated NOS is suggested. In addition to the aforementioned behavioural concerns in murine models of nNOS inhibitors, there have been concerns about multi-system effects. Dose-dependent acute kidney injury was seen in the NOSTRA trial, a study investigating the use of ronopterin in traumatic brain injury [186]; however, a subsequent RCT described this effect as only a mild, transient reduction in renal perfusion and glomerular function [187].

## 7. Conclusions, Limitations, and Future Directions

The vasoactive properties, and direct cardiac and cerebral effects of NO have long made it an attractive therapeutic target within these systems. As our understanding of the roles of NOS and its isoforms within the cardiovascular system has grown, the distinct roles played by eNOS, nNOS, and iNOS in the pathophysiology of major diseases has become clearer (Figure 1). Strategies to improve NOS activity and thereby NO signalling continue to be developed for use in conditions with a pathological lack of NO activity. Furthermore, disorders causing significant mortality and morbidity, including HFpEF and post-stroke excitotoxicity, now have evidence supporting distinct roles of the NOS isoforms in their pathophysiology, implying the possibility of selective NOS modulation. Future, studies should continue to elucidate the potential of modulating NOS activity, with a particular focus on deepening our understanding of anatomical and functional differences in NOS isoforms and harnessing this in the therapeutic approach to cardiovascular disease.

The primary limitation of this review is that we did not perform a systematic review of the topic. Nevertheless, by basing a non-systematic review of the literature on expertise of the authors, we were able to identify key articles in this field and synthesise them.

## Figures and Tables

**Figure 1 ijms-24-15200-f001:**
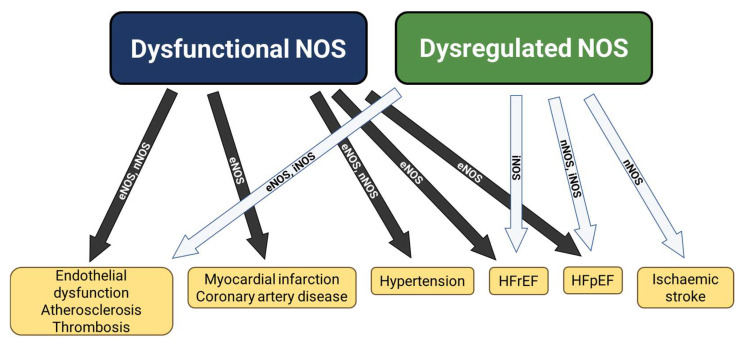
Overview of conditions with evidence for dysfunctional and dysregulated NOS activity. HFpEF: heart failure with preserved ejection fraction; HFrEF: heart failure with reduced ejection fraction.

**Figure 2 ijms-24-15200-f002:**
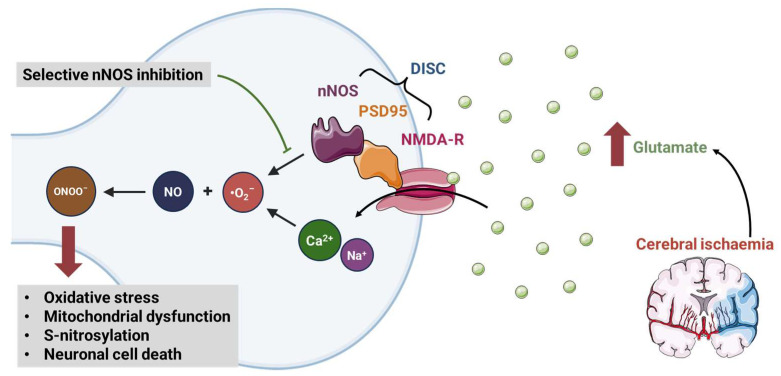
Mechanism of nNOS inhibition in post-stroke glutamate excitotoxicity: Cerebral ischaemia leads to an increase in release of the excitatory neurotransmitter glutamate. Glutamate activates the N-methyl-D-aspartate receptor (NMDA-R), with subsequent influx of calcium (Ca^2+^) (and sodium, Na^+^). Excess Ca^2+^ within the neuron contributes to the generation of superoxide (•O_2_^−^), which reacts with nitric oxide (NO) to generate peroxynitrite (ONOO^−^). Furthermore, neuronal nitric oxide synthase (nNOS) is bound to the NMDA-R by the postsynaptic density protein 95 (PSD95), forming a death-inducing signalling complex (DISC). This cascade leads to neuronal cell death via a number of mechanisms, including increased oxidative stress, mitochondrial dysfunction, and S-nitrosylation. Selective nNOS inhibition may provide a mechanism to interrupt this detrimental signalling cascade. Figure was partly generated using Servier Medical Art, provided by Servier, licensed under a Creative Commons Attribution 3.0 unported license.

## Data Availability

No new data were created or analysed in this study. Data sharing is not applicable to this article.

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
