# Peer review of "Dysfunctional and Dysregulated Nitric Oxide Synthases in Cardiovascular Disease: Mechanisms and Therapeutic Potential"

_ijms, 2023, doi:10.3390/ijms242015200_

Round 1

Reviewer 1 Report

The authors (Roy et al., Dysfunctional and dysregulated nitric oxide synthases in cardiovascular disease: mechanisms and therapeutic potential dysfunctional and dysregulated nitric oxide synthases in cardiovascular disease: mechanisms and therapeutic potential) have produced an interesting and synthetic review (state of the art) on nitric oxide in cardiovascular disease. The review includes 174 references and 2 figures. It is interesting to try to distinguish between dysfunction and dyregulation of nitric oxide synthase, this has been done here. Molecular mechanisms of action and therapeutic uses are well presented and discussed. The parallel between cardio and neuro actions is interesting to be distinguish. 

There are a few ways to improve it. 

1/ in figure 1, the abbreviations HFrEF and HFpEF should be defined in the legend of this figure.

2/ p3: KmO2 are defined for all 3 isoforms, but staining is only defined for one of them. More explanation should be given as to the significance of these data.

3/ p4 for kidney ref 40. the presence of nNOS is confirmed for each cell type in this organ ?

4/ the history of NO in vascular physiology involves EDRF and EDCF, this assessment is not presented.

5 / paragraph 5.3. ischemia/reperfusion, what is the situation during cardiac hypoxia and preconditioning on NO and NOS ?

6/ p7 some comments deal with diabetes, what is the situation within diabetic cardiomyopathy ?

7/ a sublingual form of trinitine has been developed for greater bioavailability. Is there a way to improve therapeutics involving NO and NO Synthases with these routes of administration?

 In conclusion, this review deserves to be published.

Reviewer 2 Report

Dysfunctional and dysregulated nitric oxide synthases in cardiovascular disease: mechanisms and therapeutic potential by Roy R et al., is an interesting work.  The work needs some additions to make it publishable. The suggestions are attached below.

1 . Please include the role of dysregulated NOS in inducing inflammation as a subsection under - 6. Dysregulated NOS in cardiovascular disease.

2 . Include a sub section regarding dysregulated NOS on mitochondrial health in section 6.

3 . The impact of dysregulated NOS on cellular anti-oxidant defense mechanism should be included in section 6.

4 . dysregulated NOS on cellular cross talk in cardiovascular tissue should be included as a subsection in section -6.

5 . Include a limitations section towards the end of conclusion section-7, and give the section title as Conclusion, limitations and future directions.

Only minor edits are required.
